

# Influence of urban pollution on the production of organic particulate matter from isoprene epoxydiols in central Amazonia

Suzane S. de Sá (1), Brett B. Palm (2), Pedro Campuzano-Jost (2), Douglas A. Day (2), Matthew K. Newburn (3), Weiwei Hu (2), Gabriel Isaacman-VanWertz[a] (4), Lindsay D. Yee (4), Ryan Thalman (5), Joel Brito[b] (6), Samara Carbone (6), Paulo Artaxo (6), Allen H. Goldstein (4), Antonio O. Manzi (7), Rodrigo A.F. Souza (8), Fan Mei (9), John E. Shilling (3,9), Stephen R. Springston (5), Jian Wang (5), Jason D. Surratt (10), M. Lizabeth Alexander (3), Jose L. Jimenez (2), Scot T. Martin[*] (1, 11)

(1) School of Engineering and Applied Sciences, Harvard University, Cambridge, Massachusetts, USA
(2) Department. of Chemistry & Biochemistry and Cooperative Institute for Research in Environmental Sciences, University of Colorado, Boulder, Colorado, USA
(3) Environmental Molecular Sciences Laboratory, Pacific Northwest National Laboratory, Richland, Washington, USA
(4) Dept. of Environmental Science, Policy, and Management, University of California, Berkeley, California, USA
(5) Brookhaven National Laboratory, Upton, New York, USA
(6) Departamento de Física Aplicada, Universidade de São Paulo, São Paulo, Brasil
(7) Instituto Nacional de Pesquisas da Amazonia, Manaus, Amazonas, Brasil
(8) Escola Superior de Tecnologia, Universidade do Estado do Amazonas, Manaus, Amazonas, Brasil
(9) Atmospheric Sciences and Global Change Division, Pacific Northwest National Laboratory, Richland, WA, USA
(10) Department of Environmental Sciences and Engineering, Gillings School of Global Public Health, The University of North Carolina at Chapel Hill, Chapel Hill, North Carolina, USA
(11) Department of Earth and Planetary Sciences, Harvard University, Cambridge, Massachusetts, USA
[a] Now at Massachusetts Institute of Technology, Cambridge, Massachusetts, USA
[b] Now at: Laboratory for Meteorological Physics (LaMP), University Blaise Pascal, Aubière, France

[*]To Whom Correspondence Should be Addressed
*E-mail: scot_martin@harvard.edu*



**Abstract**

2        The atmospheric chemistry of isoprene contributes to the production of a substantial mass

fraction of the particulate matter (PM) over tropical forests. Isoprene epoxydiols (IEPOX)
produced in the gas phase by the oxidation of isoprene under $HO_2$-dominant conditions are
subsequently taken up by particles, thereby leading to production of secondary organic PM. The
present study investigates possible perturbations to this pathway by urban pollution. The
measurement site in central Amazonia was located 4 to 6 hours downwind of Manaus, Brazil.
Measurements took place from February through March 2014 of the wet season, as part of the
GoAmazon2014/5 experiment. Mass spectra of organic PM collected with an Aerodyne Aerosol
Mass Spectrometer were analyzed by positive-matrix factorization. One resolved statistical
factor ("IEPOX-SOA factor") was associated with PM production by the IEPOX pathway.
Loadings of this factor correlated with independently measured mass concentrations of tracers of
IEPOX-derived PM, namely $C_5$-alkene triols and 2-methyltetrols ($R = 0.96$ and $0.78$,
respectively). Factor loading, as well as the ratio of the factor loading to organic PM mass
concentration, decreased under polluted compared to background conditions. For the study
period, sulfate concentration explained 37% of the variability in the factor loading. After
segregation of the data set by $NO_y$ concentration, the sulfate concentration explained up to 75%
of the variability in factor loading within the $NO_y$ subsets. The sulfate-detrended IEPOX-SOA
factor loading decreased by two- to three-fold for an increase in $NO_y$ concentration from 0.5 to 2
ppb. The suppressing effects of elevated NO dominated over the enhancing effects of higher
sulfate with respect to the production of IEPOX-derived PM. Relative to background conditions,
the Manaus pollution contributed more significantly to $NO_y$ than to sulfate. In this light,
increased emissions of nitrogen oxides, as anticipated for some scenarios of Amazonian
economic development, could significantly alter pathways of PM production that presently
prevail over the tropical forest, implying changes to air quality and regional climate.



## 1. Introduction

Organic compounds comprise up to 90% of the mass concentration of submicron organic particulate matter (PM) over tropical forests (Kanakidou et al., 2005). Submicron PM has adverse effects on human health (Nel, 2005; Pope III and Dockery, 2006) and influences air quality and climate by scattering radiation and acting as cloud condensation nuclei (Ramanathan et al., 2001; Kaufman et al., 2002). A significant fraction of the submicron organic material originates from secondary processes, mainly by the atmospheric oxidation of volatile organic compounds (VOCs) emitted as part of natural and human activities (Zhang et al., 2007; Hallquist et al., 2009; Jimenez et al., 2009). The particle life cycle over Amazonia is in particular strongly influenced by secondary processes that produce organic PM (Martin et al., 2010a; Pöschl et al., 2010). Biogenic emissions from tropical forests are high, and environmental conditions favor photooxidation reactions. The reactive chemistry and the relative importance of pathways leading to PM production can be strongly guided by regulating species, such as sulfate and nitric oxide (Surratt et al., 2007a; Worton et al., 2013; Liu et al., 2016a). The concentrations of these species depend on their background occurrence, pollution sources, and the relative mix of background and polluted air masses.

Over tropical forests such as Amazonia, the atmospheric chemistry of isoprene produces a substantial fraction of the submicron organic PM (Chen et al., 2009; Robinson et al., 2011; Chen et al., 2015; Isaacman-VanWertz et al., 2016). Isoprene (2-methyl-1,3-butadiene, $C_5H_8$) is the non-methane VOC most abundantly emitted by tropical forests (Guenther et al., 2012), and isoprene epoxydiols (IEPOX) have been identified as important intermediates in the production of PM from isoprene (Paulot et al., 2009; Surratt et al., 2010; Lin et al., 2012). A chemical sequence for the production of IEPOX-derived PM from the photooxidation of isoprene in the



atmosphere is represented in Figure 1. The sequence is initiated when isoprene peroxy radicals
(ISOPOO) are produced in the gas phase by reactions between isoprene and photochemically
generated hydroxyl radicals (OH). The reactive fate of the ISOPOO radicals can differ under
background compared to polluted conditions (Surratt et al., 2010; Crounse et al., 2011; Worton et
al., 2013).

Under background conditions, meaning that $HO_2$ pathways are favorable in the absence

of extensive NO pollution (Wennberg, 2013; Liu et al., 2016a), the ISOPOO radicals continue in
large part through the series of species highlighted in yellow in Figure 1. Through $HO_x$-
facilitated reaction steps, the ISOPOO radicals produce hydroperoxides (ISOPOOH) as major
first-generation products and subsequently isoprene epoxydiols (IEPOX) as major second-
generation products (Carlton et al., 2009; Paulot et al., 2009; Liu et al., 2013; St. Clair et al.,
2015; Liu et al., 2016a). Some of the produced IEPOX undergoes reactive uptake to particles, as
facilitated by hydronium ions at the surface (Surratt et al., 2007a; Lin et al., 2012; Gaston et al.,
2014; Kuwata et al., 2015; Lewandowski et al., 2015). This chemical sequence can contribute a
significant fraction of submicron PM mass concentration over tropical forests (Claeys et al.,
2004; Hu et al., 2015). Laboratory studies indicate that about half of the PM produced by
isoprene photooxidation under $HO_2$-dominant conditions in the presence of acidic sulfate
particles is associated with IEPOX production and uptake (Liu et al., 2015). Interaction of
IEPOX with cloud waters warrants investigation (Lim et al., 2005; Ervens et al., 2011;
Budisulistiorini et al., 2015; Chen et al., 2015). In addition to IEPOX pathways, laboratory
studies suggest that multifunctional hydroperoxides produced in the gas phase can contribute to
isoprene-derived PM production (Krechmer et al., 2015; Liu et al., 2016b; Riva et al., 2016b).





After reactive uptake of IEPOX, particle-phase reactions can produce several different

families of species. These species are collectively labeled "IEPOX-derived PM" and represent a
subset of the ambient organic PM, as labeled in Figure 1. The presence of 2-methyltetrols, $C_5$-
alkene triols, 3-methyltetrahydrofuran-3,4-diols, organosulfates, and related oligomers in
ambient PM is an indicator of PM production by IEPOX uptake under atmospheric conditions
(Claeys et al., 2004; Surratt et al., 2006; Surratt et al., 2007b; Surratt et al., 2010; Robinson et al.,
2011; Lin et al., 2012; Lin et al., 2014). Even though these species may differ in some cases from
the actual compounds in the atmospheric PM due to thermal decomposition during analysis
(Lopez-Hilfiker et al., 2016), they serve as chemical tracers for the atmospheric concentration of
IEPOX-derived PM (Hu et al., 2015; Isaacman-VanWertz et al., 2016). The analytical methods
highlighted in Figure 1, including that of the "IEPOX-SOA factor" of the AMS analysis used
herein, can lead to over- and underestimated IEPOX-derived PM concentrations. This
uncertainty is represented in the figure by the brown dashed lines that approximately but not
exactly correspond to IEPOX-derived PM.

Under polluted conditions, the reactive sequence of isoprene and ultimately PM

production can become significantly altered (Figure 1). NO concentrations can be sufficiently
high that ISOPOO radicals react almost entirely with NO in place of $HO_2$, thereby largely
producing methacrolein (MACR) and methyl vinyl ketone (MVK) in place of ISOPOOH (Liu et
al., 2016a). As a result, IEPOX production can be greatly decreased, ultimately reducing PM
production by IEPOX pathways. A minor channel along the NO pathway can still produce
IEPOX, although much less efficiently (Jacobs et al., 2014). Under NO-dominant conditions,
alternative pathways of PM production can become active, though in lower yields. MACR can be
oxidized to produce peroxymethylacrylic nitric anhydride (MPAN), which is a precursor to





methacrylic acid epoxide (MAE) and hydroxymethylmethyl-α-lactone (HMML), and these
compounds can undergo reactive uptake to produce PM (Kjaergaard et al., 2012; Lin et al., 2013;
Worton et al., 2013; Nguyen et al., 2015). Glyoxal produced from isoprene oxidation can
contribute to PM production (Volkamer et al., 2007; Ervens and Volkamer, 2010; McNeill et al.,
2012; Marais et al., 2016).

Another possible mechanism affecting PM production by IEPOX uptake under polluted

conditions is altered particle composition, especially particle acidity, largely driven by sulfate.
Laboratory studies show that IEPOX uptake increases with increasing acidity (Gaston et al.,
2014; Kuwata et al., 2015; Liu et al., 2015). A proposed reaction during uptake is the acid-
catalyzed ring opening of the IEPOX molecule (Surratt et al., 2010). The subsequent particle-
phase reactions include the addition of available nucleophiles, such as water to produce tetrols or
sulfate to produce organosulfates as well as their oligomers (Surratt et al., 2010; Lin et al., 2014;
Nguyen et al., 2014). In support of this proposed mechanism, analyses by positive-matrix
factorization (PMF) of mass spectra collected in the southeastern USA identified PMF factors
associated with IEPOX-derived PM, and these factors correlated positively with sulfate mass
concentrations (Budisulistiorini et al., 2013; Hu et al., 2015; Xu et al., 2015). In short, different
regimes of NO:HO$_2$ concentration ratios and different possible PM compositions between
polluted and background conditions can lead to different product distributions and different
production rates of IEPOX-derived PM.

The extent to which pollution may shift the production pathways of IEPOX-derived PM

over tropical forests remains to be elucidated. The study described herein is based on data sets
collected in the wet season downwind of Manaus, Brazil, during the *Observations and Modeling*
*of the Green Ocean Amazon* Experiment (GoAmazon2014/5) (Martin et al., 2016c). The research



site was influenced at times and to variable extents by the pollution outflow from the Manaus
metropolitan area. Compared to the background environment in Amazonia, the Manaus plume
had high number concentrations of particles and enhanced concentrations of pollutants, including
oxides of nitrogen and sulfate (Kuhn et al., 2010; Martin et al., 2016c). The reactive gas-phase
chemistry was strongly guided by the relative mix of background and polluted air masses (Trebs
et al., 2012; Liu et al., 2016a). The analysis herein focuses on how the pollution perturbed
IEPOX-derived PM production relative to background conditions.
**2. Methodology**

Data sets were collected at the "T3" site (3.2133 °S, 60.5987 °W) located 70 km to the

west of Manaus, Brazil, in central Amazonia (Martin et al., 2016c). The site was situated in a
pasture (2.5 km × 2 km) surrounded by forest. The analysis herein focuses on data sets collected
during the wet season period of February 1 to March 31, 2014, corresponding to the first
Intensive Operating Period (IOP1) of the GoAmazon2014/5 experiment.

A High-Resolution Time-of-Flight Aerosol Mass Spectrometer (HR-ToF-AMS, hereafter

AMS; Aerodyne, Inc., Billerica, Massachusetts, USA) recorded the primary data set of this
study. The AMS provided quantitative bulk characterization of atmospheric PM at a time
resolution of minutes. The design principles and capabilities of this instrument are described in
the literature (DeCarlo et al., 2006; Canagaratna et al., 2007). The instrument was housed within
a temperature-controlled research container, and the inlet to the instrument sampled from 5 m
above ground level. Detailed aspects of AMS operation are presented in the Supplement (Section
S1). In brief, ambient measurements were obtained for every 4 of 8 min. Organic, sulfate,
ammonium, nitrate, and chloride PM mass concentrations were obtained from "V-mode" data.
The choice of ions to fit was aided by the "W-mode" data, which were collected once every five



days. Data analysis was performed using *SQUIRREL* (1.56D) and *PIKA* (1.14G) of the AMS
software suite.

Positive-matrix factorization was applied to the time series of the organic component of

the high-resolution mass spectra (Ulbrich et al., 2009). The present study focuses on one of the
resolved statistical factors, referred to as the "IEPOX-SOA factor" (Hu et al., 2015). Diagnostics
of the PMF analysis, especially as related to the resolved IEPOX-SOA factor, are presented in
the Supplement (Section S2). A separate account is forthcoming to present the other PMF factors
(de Sá, in preparation). Herein, factor profile and loading refer to the mathematical products of
the multivariate statistical analysis, whereas mass spectrum and mass concentration refer to
measurements.

In complement to the AMS data sets, mass concentrations of molecular and tracer organic

species were measured using a Semi-Volatile Thermal Desorption Aerosol Gas Chromatograph
(SV-TAG) at a time resolution of one hour. The instrument collected gas and particle samples,
followed by thermal desorption, derivatization, and gas chromatography coupled to mass
spectrometry (Isaacman et al., 2014). A summary of operational details for GoAmazon2014/5 is
presented in the Supplement (Section S1), and the main account is presented in Isaacman-
VanWertz et al. (2016).

Additional data sets used in the analysis were collected at the T3 site by instruments

housed in the research container of the Mobile Aerosol Observing System (MAOS) of the ARM
Climate Research Facility (ACRF) operated by the USA Department of Energy (Mather and
Voyles, 2013; Martin et al., 2016c). A temperature-controlled inlet was mounted at 10 m above
ground level. Measurements of nitrogen oxides were made using a chemiluminescence-based
instrument (Air Quality Design). The measured odd-nitrogen family "$NO_y$", meaning $NO_x$ +





reservoir species, included NO, $NO_2$, $HNO_3$, organonitrates, particle nitrate, and peroxyacetyl
nitrates. Further details of the $NO_y$ measurements are presented in the Supplement (Section S1).
Ozone concentrations were measured by an Ozone Analyzer (Thermo Fisher, model 49i).
Particle number concentrations were measured by a Condensation Particle Counter (TSI, model
3772). Meteorological variables provided by the ARM Mobile Facility (AMF-1), which was also
part of the ACRF, included wind direction, solar irradiance, and precipitation rate. Measurements
of $NO_y$ and particle number concentration onboard the G-1 aircraft of the ARM Aerial Facility
(AAF) were also used in the analysis (Schmid et al., 2014; Martin et al., 2016c).
**3. Results and Discussion**

The organization of the presentation herein is as follows. The factor obtained from AMS

PMF analysis is presented (Section 3.1), a case study comparing background to polluted
conditions is discussed (Section 3.2), the roles of sulfate (Section 3.3) and nitric oxide (Section
3.4) in affecting factor loading are explored, and the influence of NO on production and loss
processes of IEPOX-derived PM is considered (Section 3.5).
**3.1 Statistical IEPOX-SOA factor**

Positive-matrix factorization was carried out on the time series of AMS organic mass

spectra. One statistical factor had a similar pattern of peak intensities as the "IEPOX-SOA
factor" identified in other studies (Figure S1) (Robinson et al., 2011; Slowik et al., 2011;
Budisulistiorini et al., 2013; Budisulistiorini et al., 2015; Chen et al., 2015; Xu et al., 2015). The
Pearson correlation coefficient $R$ between this factor and one obtained for a data set in the 2008
wet season in central Amazonia as part of the AMAZE-08 experiment was 0.99 (Chen et al.,
2015). The ratio $f$ of the factor loading to the mass concentration of submicron organic PM for
the present study was $0.17 \pm 0.09$ (mean ± standard deviation). The IEPOX-SOA factor has been





identified previously over the maritime tropical forest of Borneo ($f = 0.23$) (Robinson et al.,
2011), in a rural area in Canada 70 km north of Toronto ($f = 0.17$) (Slowik et al., 2011), across
several locations in the summertime southeastern USA ($f = 0.17$ to $0.41$) (Budisulistiorini et al.,
2013; Budisulistiorini et al., 2015; Hu et al., 2015; Xu et al., 2015; Budisulistiorini et al., 2016;
Marais et al., 2016), and in AMAZE-08 ($f = 0.34$) (Chen et al., 2015).

The IEPOX-SOA factor reported herein had prominent peaks at $m/z$ 53.04 and $m/z$ 82.04

(Figure S1). The ion at $m/z$ 82.04, corresponding to $C_5H_6O^+$, has been attributed to 3-methylfuran
(3-MF). The thermal degradation of isoprene-derived PM upon mass spectral analysis was
suggested as the source of 3-MF (Robinson et al., 2011). Lin et al. (2012) proposed that
sequential dehydrations upon mass spectral analysis of 3-methyltetrahydrofuran-3,4-diols, which
are an identified component of IEPOX-derived PM, can produce 3-MF. Other IEPOX-derived
species as well as non-IEPOX species might also contribute to the production of $C_5H_6O^+$ ions
(Surratt et al., 2010; Hu et al., 2015; Liu et al., 2016c).

Laboratory studies show that a mass spectrum having a pattern of peak intensities similar

to that of the IEPOX-SOA factor is produced both by the uptake of IEPOX into aqueous acidic
sulfate particles as well as by the photooxidation of isoprene under $HO_2$-dominant conditions in
the presence of acidic sulfate particles (Budisulistiorini et al., 2013; Nguyen et al., 2014; Kuwata
et al., 2015; Liu et al., 2015). The possibility of similar uptake by a broader range of liquid media
remains to be fully tested, such as other acidic solutions as well as cloud waters. Compared to the
laboratory spectra of (Liu et al., 2015),  representing about 4 h of OH exposure at atmospheric
concentrations ($1.7 \times 10^6$ molec cm$^{-3}$), the main difference was the relative intensity of the $m/z$
44 peak. For the IEPOX-SOA factor of the present study, this peak was four times more intense



(Figure S1), suggesting that the atmospheric PM was more oxidized. Hu et al. (2016) showed
that heterogeneous aging of IEPOX-SOA can result in increased relative signal at $m/z$ 44.
By contrast, laboratory studies show that a significantly different mass spectrum from
that of the IEPOX-SOA factor is obtained for PM produced from isoprene photooxidation in the
absence of aqueous particles (Krechmer et al., 2015; Kuwata et al., 2015). Under these
conditions, chemical pathways other than IEPOX uptake into a liquid medium appear to be
active, such as the condensation of low-volatility, multifunctional compounds produced by
additional oxidation of ISOPOOH (Krechmer et al., 2015; Liu et al., 2016b; Riva et al., 2016b).
This non-IEPOX pathway, however, is not expected to contribute a large fraction of the
produced PM during the study period because of the high RH conditions in Amazonia and the
prevalence of liquid particles for the prevailing atmospheric conditions (Bateman et al., 2016; de
Sá, in preparation).
The SV-TAG measurements of the concentrations of $C_5$-alkene triols and 2-methyltetrols
support the interpretation of the IEPOX-SOA factor as an indicator that PM was being produced,
at least in significant part, from the reactive uptake of IEPOX (Claeys et al., 2004; Wang et al.,
2005; Surratt et al., 2010). The factor loading strongly correlated with the concentrations of $C_5$-
alkene triols ($R = 0.96$) and 2-methyltetrols ($R = 0.78$) (Figure 2). These species have been
associated with the IEPOX reaction pathway in several laboratory studies (Surratt et al., 2010;
Riedel et al., 2016). The $R$ value with respect to $C_5$-alkene triols was independent of the $f_{peak}$
value of the PMF solution, demonstrating the robustness of the relative time trend of factor
loading even though the factor profile and absolute loadings changed across $f_{peak}$ values (Figure
S2d).





The loading of the IEPOX-SOA factor may be an overestimate or an underestimate of the
atmospheric concentration of the IEPOX-derived PM (Supplement, Section S2). The IEPOX-
SOA factor can be understood as the net result of (i) produced IEPOX-derived PM, (ii) less that
portion of the carbon that gets further oxidized and mixed into other PMF factors, and (iii) plus
that portion of non-IEPOX-derived PM that gives rise to a similar AMS mass spectral pattern as
the IEPOX-derived PM (Supplement, Section S2). Processes of type ii contribute to
underestimates and processes of type iii lead to overestimates when using IEPOX-SOA factor
loading as a surrogate for IEPOX-derived PM concentration. These uncertainties are
qualitatively represented in Figure 1 by the brown dashed lines that enclose the fraction of
particle material statistically captured by the factor analysis. The further analysis herein is based
on using the loading of the IEPOX-SOA factor as a scalar proxy for the mass concentration of
IEPOX-derived PM in a sampled air mass.
**3.2 Background compared to polluted conditions**
Under background conditions in the wet season, remote areas of the Amazon forest
constitute one of the least polluted continental regions on Earth (Martin et al., 2010a). Nitric
oxide (NO) concentrations characteristic of central Amazonia range from 20 to 70 ppt (Torres
and Buchan, 1988; Bakwin et al., 1990; Levine et al., 2015). Daytime maximum ozone
concentrations are 10 to 15 ppb (Rummel et al., 2007). Sulfate mass concentrations associated
with in-basin processes are on average $< 0.1$ $\mu g\ m^{-3}$, and total background sulfate concentrations
contributed by in- and out-of-basin processes rarely exceed 0.5 $\mu g\ m^{-3}$ (Andreae et al., 1990;
Chen et al., 2009).
In the wet season, Manaus emissions were the most important anthropogenic influence on
observations at the T3 research site (Martin et al., 2016a). The afternoons of March 3 and 13,





2014, are presented herein as representative cases of background and polluted conditions,
respectively. Both days were sunny, and major precipitation events were absent. Particle number
concentrations measured onboard the G-1 aircraft within the atmospheric boundary layer show
the position of the pollution plume on these two afternoons (Figure 3). $NO_y$ concentrations
measured during the same flight are shown in Figure S3. The visualization in Figure 3 shows that
on March 3 the Manaus plume passed south of the T3 site. By comparison, on March 13 the
central portion of the plume passed over T3. Aircraft-based observations to track the Manaus
plume were available for 16 afternoons of the two-month study period. Ground site diagnostics
of the urban pollution reaching the T3 site were therefore also needed.

Measurements at ground level at the T3 site are plotted in Figure 4 for the afternoons of

March 3 (left panel) and March 13 (right panel). Based on wind speeds, the research site was 4 to
6 h downwind of Manaus (Martin et al., 2016c). Anthropogenic-biogenic interactions affecting
the production of IEPOX-derived PM were driven in large part by atmospheric photochemistry
at daybreak. Morning urban emissions followed by atmospheric processing arrived at the T3 site
during the local afternoon. The afternoon period, in addition to the connection to the Manaus
plume, was also characterized by reduced variability in other possible confounding variables,
such as temperature, radiation, and relative humidity. Figure 4 shows that on the afternoon of
March 3 ozone concentrations were below 10 ppb, particle number concentrations were below
1000 $cm^{-3}$, $NO_y$ concentrations were less than 1 ppb, and sulfate concentrations were 0.3 to 0.4
$\mu g\ m^{-3}$. Species concentrations were stable throughout the afternoon. On March 13, ozone
concentrations exceeded 30 ppb for most of the afternoon, particle concentrations reached 10,000
$cm^{-3}$, $NO_y$ concentrations consistently exceeded 1 ppb, and sulfate concentrations were 0.3 to 0.6



$\mu g\ m^{-3}$. Concentrations fluctuated markedly throughout the afternoon on March 13, reflecting
different levels of pollution influence in the air passing over the T3 site during that period.

Elevated concentrations of ozone, particle number, and $NO_y$ were reliable markers of

pollution influence over the course of the study period (Supplement, Section S3). Pollution was
associated with stronger relative enhancements in $NO_y$ concentrations than in sulfate
concentrations (Sections 3.3 and 3.4). With respect to the IEPOX-SOA factor, Figure 4 shows
that the absolute and relative loadings decreased for the polluted compared to background
conditions. Relative loadings are expressed by the ratio $f$ of IEPOX-SOA factor loading to the
organic PM mass concentration. Decreased absolute and relative factor loadings under polluted
conditions, presented in Figure 4 as a case study, also characterized the data sets of the entire
study period. Other examples are presented in the Supplement (Figure S4).
**3.3 Sulfate as a driver of IEPOX-derived PM production**

A scatter plot between sulfate mass concentrations and IEPOX-SOA factor loadings for

all afternoon periods is shown in Figure 5a. Background and polluted conditions are represented
in the data set. For further visualization, the data set was organized into six subsets based on
sulfate concentration. The medians and the means of the subsets are plotted in the figure. The
visualization shows that sulfate concentration served as a first-order predictor of the IEPOX-
SOA factor loading in central Amazonia in the wet season. The explanation can be a
combination of increased acidity, greater reaction volume including by enhanced hygroscopic
growth, and possibly a nucleophilic role for sulfate (Xu et al., 2015; Marais et al., 2016). An
analysis of the relative importance of each is out of the scope of the present study (Supplement,
Section S4).



For Figure 5a, the coefficient $R^2$ of determination between sulfate mass concentration and

factor loading was 0.37, meaning that 37% of the variance of the IEPOX-SOA factor loading
was explained by sulfate mass concentration. As a point of comparison, $R^2$ varied between 0.4
and 0.6 for observations in the southeastern USA, which seasonally is a region of high isoprene
emissions (Budisulistiorini et al., 2013; Budisulistiorini et al., 2015; Hu et al., 2015; Xu et al.,
2015). A chemical transport model that predicted IEPOX-derived PM mass concentrations for
the southeastern USA obtained $R^2$ of 0.4 for the relationship to predicted sulfate mass
concentration (Marais et al., 2016). The model attributed the correlation to the acidity and
particle volume provided by sulfate, both of which favored IEPOX uptake. Central Amazonia
and the southeastern USA differ considerably in terms of meteorology, chemistry, and levels of
regional pollution, yet they have in common an important role of sulfate concentration as a
predictor of IEPOX-derived PM concentration, even as the sulfate concentrations themselves
differ by an order of magnitude. Sulfate concentrations typically had an interquartile range of
[1.5, 3.0] $\mu$g m$^{-3}$ in the studies in the southeastern USA, which can be compared to a range of
[0.11, 0.36] $\mu$g m$^{-3}$ under background conditions during the wet season in central Amazonia.

A key difference between the southeastern USA and central Amazonia is the role of

sulfate concentration as a clear or ambiguous indicator, respectively, of urban influence. For the
relatively low sulfate mass concentrations (<0.5 $\mu$g m$^{-3}$) characteristic of the study period,
background air in central Amazonia contributed significantly to the variability in sulfate
concentration measured at the T3 site. Background concentrations of sulfate in Amazonia,
distinguished from sulfate tied to the urban Manaus plume, originated from in-basin emissions of
dimethyl sulfide (DMS) and hydrogen sulfide (H$_2$S) from the forest as well as from out-of-basin
marine emissions from the Atlantic Ocean (Andreae et al., 1990; Chen et al., 2009; Martin et al.,




2010a). In the wet season, biomass burning from Africa and to a lesser extent from South
America also episodically contributed significantly to sulfate concentrations in the Manaus
region. In addition, emissions from large cities on the eastern coast of Brazil were important at
times when rare meteorological events shifted the northeasterlies typical of the wet season to
easterlies (Martin et al., 2016a). Manaus contributions to sulfate mass concentrations in an air
mass were in addition to these various background sources.

The relative importance of Manaus contributions to the sulfate concentrations in the air

masses that passed over T3 was assessed by comparison of the probability density function of
sulfate concentration at T3 to those of sites upwind of Manaus (Figure 5b). The distributions of
the two upwind sites had a central tendency of 0.05 to 0.3 $\mu g\ m^{-3}$, suggesting the range of natural
concentrations, and a rightside skewness up to 0.6 $\mu g\ m^{-3}$, suggesting the importance of episodic
long-range transport (Chen et al., 2009). The figure shows that the distribution at T3 did not
differ greatly from those of the upwind sites even though the air masses over T3 regularly
transported Manaus pollution, indicating that Manaus did not constitute a dominant sulfate
source in the region. Elevated sulfate concentrations on any one afternoon at the T3 site might
have arisen because of elevated background concentrations on that day rather than the influence
of the Manaus pollution plume. The implications are that (i) sulfate concentration was an
ambiguous indicator of urban influence at the T3 site and (ii) increases in sulfate concentrations
in pollution events were moderate relative to background concentrations.
**3.4 NO as a modulator of IEPOX-derived PM production**

In the transport from Manaus to the T3 research site, NO concentration is not conserved,

in part because of reactions with ozone and organic peroxy radicals(Martin et al., 2016a). In this
case, the instantaneous NO concentrations measured at the T3 site do not directly provide



information about the fate of ISOPOO radicals along the transport time of 4 to 6 h from Manaus
to the T3 site. The collective contributions of NO, $NO_2$, and their oxidation products are,
however, reflected in measurements of $NO_y$ concentrations at the T3 site. The $NO_y$ family is
expected to have a longer lifetime than the transport time from Manaus to the T3 site (Romer et
al., 2016). The $NO_y$ concentration measured at T3 therefore served as a surrogate for the
integrated exposure of the airmass to NO chemistry between Manaus and T3 (Liu et al., 2016a).

Unlike the ambiguity associated with the sulfate concentration, an elevated $NO_y$

concentration served as a clear indicator of anthropogenic influence in an air mass passing over
the T3 site. For background conditions over the forest, $NO_y$ originated from NO emitted from
soils and other natural sources such as lightning (Bakwin et al., 1990; Jacob and Wofsy, 1990).
The probability density function of $NO_y$ concentration under background conditions in the wet
season of the central Amazon basin is shown in Figure 6b (Bakwin et al., 1990). The distribution
for measurements at T3 is also shown. Relative to the narrow distribution around 0.5 ppb for
background conditions, there is high-side skewness extending up to 4 ppb for the T3
measurements, indicating the clear influence of Manaus emissions on $NO_y$ concentrations.

$NO_y$ concentration was incorporated into the analysis by segregation of the dataset of

Figure 5a into five subsets (Supplement, Section S5). Linear fits to the $NO_y$-segregated data
subsets are plotted in Figure 6a. Each subset is represented by a different color. Parameter values
of the associated fits are listed in Table 1. In conjunction with sulfate concentration, the
visualization presented in Figure 6a shows that $NO_y$ concentration further explained the
variability in IEPOX-SOA factor loadings. The $R^2$ values, representing the extent to which
sulfate was able to explain variability in IEPOX-SOA factor loading once isolated for $NO_y$
concentration, were higher for the data subsets having lower and higher extremes of $NO_y$



concentrations (Table 1). These conditions represent the limiting cases of fully background
conditions for the former and the strongest effects of Manaus pollution for the latter. By
comparison, intermediate $NO_y$ concentrations could arise from air masses that mixed together
background air with Manaus pollution during the transport to T3 (e.g., by entrainment) and thus
represent complex processing. Single or multiple mixing points could occur anywhere along the
path from Manaus to T3, thus introducing variability into the effective photochemical age of the
air mass arriving at T3 and resulting in lower $R^2$ values for intermediate $NO_y$ concentrations. In
caveat, this explanation assumes that NO emissions from Manaus had low day-to-day variability.

In relation to the influence of Manaus pollution, sulfate concentration was affected by a

mixture of background and urban sources whereas $NO_y$ concentration largely had urban sources.
As an approximation to keeping the sulfate concentration constant and thus focusing on the role
of NO in the urban pollution, the visualization of the dependence of IEPOX-SOA factor loading
on $NO_y$ concentration was further refined by taking data subsets segregated by low (< 0.1 μg
$m^{-3}$) and high (> 0.3 μg $m^{-3}$) sulfate concentrations. Figures 7a, 7b, and 7c show the factor
loading, organic PM mass concentration, and the ratio $f$ of the IEPOX-SOA factor loading to the
organic PM mass concentration, respectively, plotted against $NO_y$ concentration for low and high
sulfate concentrations.

Figure 7a shows that for both low and high sulfate concentrations an increase in $NO_y$

concentration from background to polluted concentrations was associated with a decrease in the
IEPOX-SOA factor loading by two to three times. For low sulfate concentration, the interquartile
range of the factor loading decreased from [0.037, 0.093] to [0.022, 0.039] μg $m^{-3}$ for an increase
in $NO_y$ concentration from 0.5 to 2 pbb. For high sulfate concentration, the factor loading
decreased from [0.57, 0.95] to [0.21, 0.35] μg $m^{-3}$ for the same transition in $NO_y$ concentration.



The greatest changes in factor loading were in the region of 1 ppb $NO_y$. This region of greatest
sensitivity coincided with the transition from background to polluted conditions. For the same
time period, a change was reported in the gas phase from a dominance of ISOPOOH to
MVK/MACR products across this transition in $NO_y$ concentration (Liu et al., 2016a).

Figure 7b shows that for both low and high sulfate concentrations the organic PM mass

concentration $M_{org}$ and the IEPOX-SOA factor loading had opposite trends for low compared to
intermediate $NO_y$ concentrations, even though the trend in $M_{org}$ was less steep. The factor
loadings decreased by 60% whereas the $M_{org}$ increased by 25% for 0.5 to 2 ppb $NO_y$ (Figures 7a
and 7b). Increases in $M_{org}$ can include contributions from secondary PM produced by enhanced
concentrations of hydroxyl radicals and ozone in the pollution plume as well as from primary
PM emitted from the Manaus urban region (Martin et al., 2016a; de Sá, in preparation). For
higher $NO_y$ concentrations (> 2 ppb), however, Figure 7b shows that $M_{org}$ decreased after a peak
value, approaching values close to background under the most polluted conditions. The
chemistry can become sufficiently shifted that more-volatile gas-phase products can be produced
(Pandis et al., 1991; Kroll et al., 2005; Carlton et al., 2009). In addition, hydroxyl radical
concentrations can also decrease because of titration by $NO_2$ (Valin et al., 2013; Rohrer et al.,
2014). An increase in total organic mass concentration could possibly contribute to a decrease in
IEPOX-derived PM production by kinetically limiting the uptake of IEPOX (Gaston et al., 2014;
Lin et al., 2014; Riva et al., 2016a). The dominant effect of the urban plume, however, seems to
be that of shifting the fate of ISOPOO radicals through the increase in NO, thereby significantly
decreasing the production of ISOPOOH (Liu et al., 2016a) (Section 3.5).

The combined trends of Figures 7a and 7b for increasing $NO_y$ are represented in Figure

7c as the ratio $f$. The figure shows that $f$ decreased for increasing $NO_y$ concentration for both low



and high sulfate concentrations. The greatest decrease occurred across the range of $NO_y$
concentrations that represented the shift from background to polluted conditions. For low sulfate
concentration, the interquartile range of $f$ decreased from [0.09, 0.18] to [0.04, 0.09] for an
increase in $NO_y$ concentration from 0.5 to 2 ppb. These ranges shifted to [0.35, 0.40] and [0.07,
0.18] for high sulfate concentration. The magnitude of the decrease for high sulfate
concentrations suggests that IEPOX-derived PM shifted from being a major to a minor
component of the PM, although the caveats related to under- and overestimates connected to the
IEPOX-SOA factor should be kept in mind (vide supra). Taken together, the results shown in
Figure 7 demonstrate how urban pollution affected the production and composition of regional
IEPOX-derived PM.

The data sets presented in Figures 5, 6, and 7 lead to the conclusion that the additional

NO concentrations contributed by Manaus emissions typically suppress the production of
IEPOX-derived PM to a greater extent than the additional sulfate concentrations enhance it.
Figure 8 presents a systematic visualization. The factor loadings at T3 are represented as
contours for axes of sulfate and $NO_y$ concentrations. Higher factor loadings are favored for
higher sulfate and lower $NO_y$ concentrations. Factor loadings are most sensitive to changing
concentration in the high-sulfate, low-$NO_y$ region. The gray dashed line in Figure 8 represents a
qualitative divisor between domains of typical background and polluted conditions downwind of
Manaus.
**3.5 Influence of NO on production and loss processes of IEPOX-derived PM**

Elevated NO may affect both the production and loss processes of IEPOX-derived PM.

On the one hand, production may be reduced because of increased scavenging of ISOPOO by
NO, thus obviating production of IEPOX and consequently of IEPOX-derived PM. Production



may also be reduced because of more rapid gas-phase loss of IEPOX in response to elevated OH
and $O_3$ concentrations. On the other hand, loss of IEPOX-derived PM may be enhanced due to
faster processing of its characteristic compounds by the elevated oxidant concentrations.

A Lagrangian model is employed to help delineate the relative importance of reduced

production compared to enhanced loss on the observed IEPOX-derived PM concentrations. The
model is initialized by background air that passes over Manaus in the mid-morning. The
evolution of IEPOX-derived PM in that air mass is modeled under either polluted or background
conditions for arrival at the T3 site in the afternoon. The governing differential equation of the
model represents the sum of production and loss processes affecting the concentrations of
IEPOX-derived PM, as follows:
$$\frac{dM}{dt} = -\alpha_L k_L M + \alpha_P k_P \tag{1}$$

where $M$ designates the IEPOX-derived PM mass concentration, $t$ designates time, and the first
and second terms on the right-hand side represent loss and production processes, respectively.
Table 2 lists other symbol definitions and units.

The analytic solution of Equation 1 for time $t$ is presented in the Supplement (Section

S6). From this solution, characteristic times $\tau$ for production and loss processes for polluted
compared to background conditions are as follows: $\tau_{P,pol} = M_0 / (\alpha_P k_P)$, $\tau_{P,bg} = M_0 / k_P$, $\tau_{L,pol} = 1 /$
$(\alpha_L k_L)$, and $\tau_{L,bg} = 1 / k_L$ (Supplement, Section S6). The term $M_0$ represents the IEPOX-derived
PM mass concentration just upwind of Manaus. Under background conditions, the enhancement
factors $\alpha_L$ and $\alpha_P$ are unity by definition. Under polluted conditions, $\alpha_L = 2$ and $\alpha_P = 0.1$ to reflect
enhanced loss and decreased production, respectively. Further descriptions of the model and
assumptions are presented in the Supplement (Section S6).



The analysis strategy is to compare $\tau_P$ and $\tau_L$ to the transport time $\tau_{tr}$ under polluted and
background conditions to assess the relative importance of altered production and loss processes
for IEPOX-derived PM from Manaus to T3. The mode value for $\tau_{tr}$ is 4 h based on trajectory
analysis (Martin et al., 2016c). Intervals for the characteristic times $\tau_P$ and $\tau_L$ are constrained by
the T3 afternoon data sets. Concentration ratios $\xi$, defined as $\xi = M_{pol} / M_{bg}$, are used to constrain
the model (Table 3). The quantities $M_{pol}$ and $M_{bg}$ denote $M(t = \tau_{tr})$, meaning the mass
concentration at T3 under polluted or background conditions, respectively. The use of the ratio
quantity $\xi$ in the analysis, rather than absolute concentrations, provides increased robustness
because of low variability in $\xi$ across the observed range of sulfate concentrations, even as $M_{pol}$
and $M_{bg}$ vary greatly (Table 3). The possible impact of over- or underestimates of IEPOX-
derived PM mass concentration, as a consequence of using IEPOX-SOA factor loading as a
surrogate, is also mitigated by the use of $\xi$.
Two cases of the model (1 and 2) are presented, respectively focusing on constraining $k_P$
or $k_L$ and consequently $\tau_P$ or $\tau_L$ (Table 4). The results for Case 1 of the analysis are shown in
Figure 9a. The value of $k_P$ is varied from 0 to 0.2 $\mu g\ m^{-3}\ h^{-1}$ while the other model parameters are
held constant. The loss rate coefficient $k_L$ is fixed at 0.015 $h^{-1}$, corresponding to a characteristic
time of 2.8 days (Supplement, Section S6). Based on observed values of $\xi$ (gray shaded area in
Figure 9a), an interval for $k_P$ of [0.07, 0.13] $\mu g\ m^{-3}\ h^{-1}$ is obtained, as indicated by the vertical
dashed lines. Across this interval, $M_{bg}$ and $M_{pol}$ vary from 0.49 to 0.72 $\mu g\ m^{-3}$ and 0.23 to 0.25 $\mu g$
$m^{-3}$, respectively, which are consistent with the observed IEPOX-SOA factor loadings (Table 3).
The modeled production times have intervals of [1.8, 3.3] h for $\tau_{P,bg}$  and [18, 33] h for $\tau_{P,pol}$.
Case 2 of the analysis evaluates constraints on the loss rate coefficient $k_L$, and results are
shown in Figure 9b. Loss processes can include chemistry, such as heterogeneous oxidation or



other in-particle reactions that reduce the IEPOX-SOA factor loading, as well as physical
mechanisms, such as particle deposition and particle dilution by entrainment that reduce mass
concentrations of IEPOX-derived PM (Supplement, Section S6). The value of $k_L$ is varied over
three orders of magnitude, representing characteristic times of hours to weeks, while the other
model parameters are held constant (Table 4). The production rate coefficient $k_P$ is fixed at 0.10
µg m$^{-3}$ h$^{-1}$, corresponding to the interval midpoint of Case 1. The observed values of $\xi$ (gray
shaded area) in intersection with the modeled values of $\xi$ imply an upper limit on $k_L$ at 0.043 h$^{-1}$,
corresponding to characteristic times of a day to weeks (Figure 9b). Correspondingly, $\tau_{L,bg,} > 24$
h under background conditions, and $\tau_{L,pol} > 12$ h under polluted conditions.

The analyses of Cases 1 and 2 constrain the values of $\tau_{P,pol}$, $\tau_{P,bg}$, $\tau_{L,pol}$, and $\tau_{L,bg}$ based on

the observed values of $\xi$. The lower limits of the characteristic times for loss, meaning $\tau_{L,bg,} > 24$
h and $\tau_{L,pol} > 12$ h, are considerably longer than the transport time of 4 h under both background
and polluted conditions. Enhanced loss, therefore, does not explain alone the observed values of
$\xi$. By comparison, the observed values of $\xi$ imply a shift in the characteristic time for production
from [1.8, 3.3] h under background conditions to [18, 33] h under pollution conditions. The shift
in timescale is significant in light of the transport time of 4 h. Therefore, reduced production,
rather than enhanced loss, is consistent with the lower IEPOX-derived PM concentrations under
polluted conditions. A few afternoon hours of altered isoprene chemistry is sufficient to
significantly shift the atmospheric concentration of IEPOX-derived PM.
**4. Summary and conclusions**

The influence of anthropogenic emissions on the production of organic particulate matter

from isoprene epoxydiols was studied during the wet season of the tropical forest in central
Amazonia. The IEPOX-derived PM concentration at the T3 site, as indicated by the IEPOX-





SOA factor loading, was lower under polluted compared to background conditions. Sulfate
concentration was an important first-order predictor of the IEPOX-SOA factor loading,
corroborating the understanding of the role of sulfate in the production of IEPOX-derived PM
that has been developed in laboratory studies as well as in investigations in the southeastern USA
(Surratt et al., 2007b; Budisulistiorini et al., 2013; Budisulistiorini et al., 2015; Hu et al., 2015;
Kuwata et al., 2015; Xu et al., 2015). Unlike the southeastern USA, however, where
anthropogenic influences dominated variability in sulfate concentrations, contributions by the
Manaus urban region to sulfate concentrations were of approximately equal magnitude to the
background variability in central Amazonia. By comparison, Manaus urban emissions of NO
dominated over background concentrations, and the $NO_y$ concentration measured 4 to 6 h
downwind of Manaus at the T3 site was an important predictor of the IEPOX-SOA factor
loading. In net effect, the suppression of IEPOX production because of elevated NO
concentrations in the pollution plume dominated over any enhancements in IEPOX uptake
because of greater sulfate concentrations.

The dependence of the IEPOX-SOA factor loadings on both sulfate and $NO_y$

concentrations, as shown in Figure 8, suggests that altered net anthropogenic effects may be
expected for different geographic regions, even within Amazonia, and different time periods,
such as the wet and dry seasons. The T3 site experienced a wide range of $NO_y$ concentrations,
allowing for the systematic demonstration of the dependence of IEPOX-derived PM
concentrations on $NO_y$ concentrations. The results show that the transition in isoprene
photochemistry related to the production of IEPOX-derived PM is most sensitive precisely at the
transition between background and polluted conditions, around 1 ppb of $NO_y$, at least for central
Amazonia in the wet season. These findings suggest that the fraction of PM derived from IEPOX



might be lower and have lower variability for other geographic regions characterized by higher
$NO_y$ baseline concentrations (e.g., upward of 1 to 2 ppb). For regions further downwind of the
urban center, the effects of the plume are expected to phase out both due to dilution and to
consumption of NO, and a gradual transition to background chemistry is expected to take place.
Adequately representing background conditions and the transition to polluted conditions within
models, including the dependence of the production of IEPOX-derived PM not only on sulfate
but also on NO concentration, is thus important for making accurate predictions of PM
concentrations, both in Amazonia and around the globe.

The findings herein can be considered in the context of Amazonia in transition (Davidson

et al., 2012). In the past 50 years, the metropolitan area of Manaus, today at more than 2 million
inhabitants, has experienced rapid economic and population growth (Martin et al., 2016c).
Changes in the fuel matrix, such as the ongoing shift from high-sulfur to low-sulfur oil in the
vehicle fleet as well as from fuel oil to natural gas in many power plants (Medeiros et al., in
preparation), are changing the composition of the Manaus pollution plume. Based on the findings
presented herein, a reduction in sulfate sources from Manaus would not be expected to
considerably affect the mass concentration of IEPOX-derived species in forest regions affected
by the plume. Background sources independent of Manaus appear sufficient to sustain sulfate
concentrations regionally. On the other hand, in the absence of pollution control technologies,
NO emissions can be expected to increase in coming years due to the development of more
efficient (i.e., higher temperature) sources of electricity associated with the development of
natural gas resources in the basin, as well as from growth in transportation associated with
increased population. Increased NO concentrations can be expected to reduce the mass
concentration of IEPOX-derived species in forest regions affected by the plume. Changes in the



atmospheric particle population can have follow-on effects on cloud type, duration, and rainfall
(Pöschl et al., 2010). In addition to PM derived from IEPOX as discussed herein, a better
understanding of other pathways that also contribute to organic PM, as well as possible changes
to those pathways with increasing pollution in the region, warrants further study so as to achieve
sufficient knowledge for decision-making related to air quality and climate in Amazonia.



**Acknowledgments.** Institutional support was provided by the Central Office of the Large Scale Biosphere Atmosphere Experiment in Amazonia (LBA), the National Institute of Amazonian Research (INPA), and Amazonas State University (UEA). We acknowledge support from the Atmospheric Radiation Measurement (ARM) Climate Research Facility, a user facility of the United States Department of Energy (DOE), Office of Science, sponsored by the Office of Biological and Environmental Research, and support from the Atmospheric System Research (ASR) program of that office. Additional funding was provided by the Amazonas State Research Foundation (FAPEAM), the São Paulo State Research Foundation (FAPESP), the USA National Science Foundation (NSF), and the Brazilian Scientific Mobility Program (CsF/CAPES). S. de Sá acknowledges support by the Schlumberger Foundation, Faculty for the Future Fellowship. The research was conducted under scientific license 001030/2012-4 of the Brazilian National Council for Scientific and Technological Development (CNPq).



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





**List of Tables**

| Group | NO$_y$ range (ppb) | Fit slope | Fit intercept | Fit $R^2$ |
|:---:|:---:|:---:|:---:|:---:|
| 1 | $< 0.66$ | 2.16 | -0.13 | 0.75 |
| 2 | $0.66 - 0.92$ | 1.48 | -0.04 | 0.64 |
| 3 | $0.92 - 1.55$ | 0.78 | 0.06 | 0.24 |
| 4 | $1.55 - 2.45$ | 0.71 | -0.01 | 0.44 |
| 5 | $> 2.45$ | 0.55 | -0.02 | 0.62 |

**Table 1.** Parameters associated with NO$_y$ groupings in Figure 6. Listed are the NO$_y$ concentrations and the parameters for least-squares linear fits to each group. $R^2$ represents the coefficient of determination.


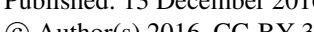


| Symbol | Description | Unit |
|---|---|---|
| $M$ | mass concentration of IEPOX-derived PM | $\mu g\ m^{-3}$ |
| $t$ | time | h |
| $k_P$ | zero-order rate coefficient for production under background conditions | $\mu g\ m^{-3}\ h^{-1}$ |
| $k_L$ | first-order rate coefficient for loss under background conditions | $h^{-1}$ |
| $\alpha$ | multiplicative factor representing the effects of Manaus pollution on rate coefficients | |
| $\tau$ | characteristic time of a process (e.g., production, loss, or transport) | h |
| Subscript $tr$ | Refers to transport | |
| Subscript $bg$ | Refers to background conditions | |
| Subscript $pol$ | Refers to polluted conditions | |
| Subscript 0 | Refers to an initial state (i.e., just upwind of Manaus) | |
| Subscript $P$ | Refers to production processes | |
| Subscript $L$ | Refers to loss processes | |

**Table 2**. Descriptions and units of symbols in the model.



| | Loading (µg m$^{-3}$) for background conditions | | Loading (µg m$^{-3}$) for polluted conditions | | Ratio $\xi$ | |
|---|---|---|---|---|---|---|
| | Low sulfate | High sulfate | Low sulfate | High sulfate | Low sulfate | High sulfate |
| IEPOX-SOA factor | [0.037, 0.093] | [0.57, 0.95] | [0.022, 0.039] | [0.21, 0.35] | 0.47 | 0.35 |

**Table 3**. Interquartile intervals of IEPOX-SOA factor loadings observed for background and polluted conditions. Background and polluted conditions correspond to approximately 0.5 ppb and 2 ppb of NO$_y$, respectively. The table also lists the resulting ratio $\xi$ of the median factor loading under polluted compared to background conditions.





| Model case | Parameter values | | | | Initial condition |
|---|---|---|---|---|---|
| | $k_P$ | $k_L$ | $\alpha_P$ | $\alpha_L$ | $M_0$ |
| 1. Vary $k_P$ | 0 to 0.2 | 0.018 | 0.1 | 3 | 0.23 |
| 2. Vary $k_L$ | 0.065 | 0.001 to 1 | 0.1 | 3 | 0.23 |

**Table 4**. Parameter values and initial conditions used in model cases. Descriptions and units are listed in Table 2. The $M_0$ value is based on the mean IEPOX-SOA factor loading that was measured from 14:00-16:00 UTC (10:00-12:00 local time) at a regional background site during two months of the wet season in 2008 (Chen et al., 2015). For comparison, a similar value of 0.19 $\mu$g m$^{-3}$ was obtained during the present study as the mean value observed at the T3 site for $NO_y < 1$ ppb (14:00-16:00 UTC).



**List of Figures**

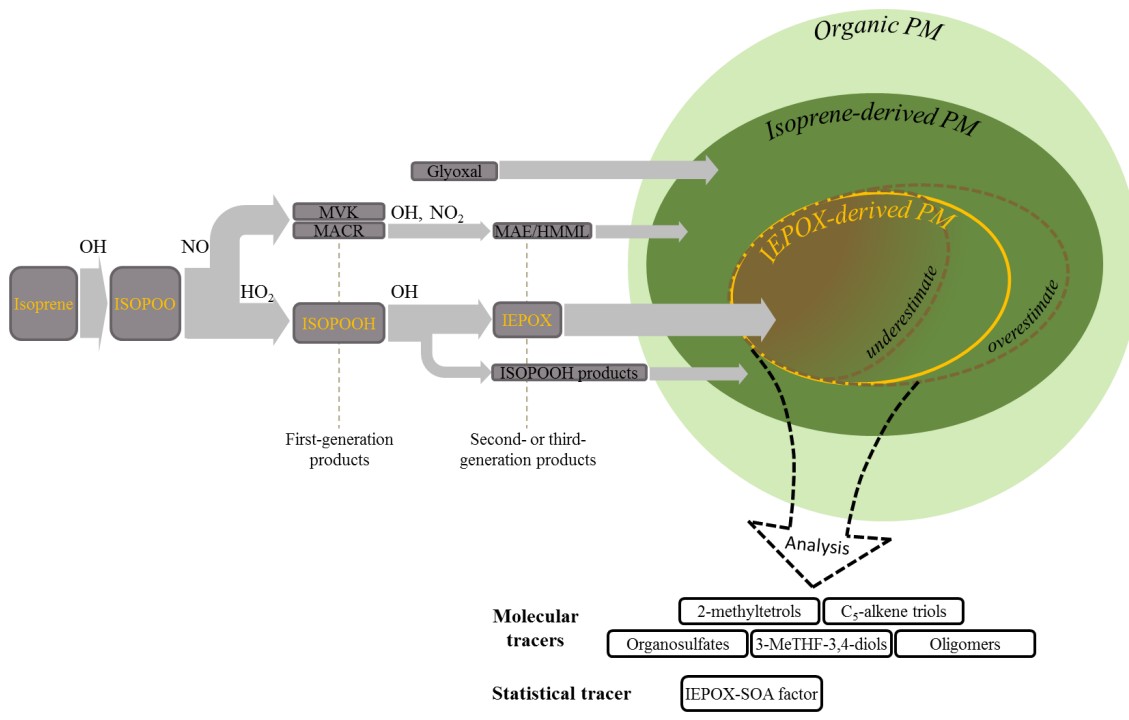

**Figure 1.** Schematic diagram for the production of IEPOX-derived PM from the photooxidation of isoprene. Organic peroxy radicals (ISOPOO), produced by OH attack and $O_2$ addition to isoprene, are scavenged along NO or $HO_2$ pathways. By the $HO_2$ pathway, organic hydroperoxides (ISOPOOH) are first-generation products that react with additional OH to produce isoprene epoxydiols (IEPOX). The IEPOX species undergo reactive uptake into particles, ultimately producing IEPOX-derived particulate matter. Arrow thickness qualitatively illustrates the relative importance (i.e., mass flux) of a reaction channel under background conditions. Gray and green background colors indicate species in the gas and particle phases, respectively. The light-green disk represents the total organic PM. Within that disk, the contribution by isoprene-derived PM, including compounds produced both IEPOX and non-IEPOX pathways, is represented by the dark-green oval. Inside that oval, the contribution by IEPOX-derived PM is represented by the yellow oval region. The color gradient between brown and dark green illustrates the chemical modification of the IEPOX-derived PM over time. The large dashed black arrow represents the analytical methods that use different types of molecular and statistical tracers (listed in the boxes) to quantify the IEPOX-derived PM mass concentrations. For simplicity, the figure omits the many routes leading to the production of glyoxal (Fu et al., 2008), possible ISOPOO isomerization when NO and $HO_2$ concentrations are sufficiently low (Crounse et al., 2011; Liu et al., 2016a), second-generation production of peroxymethylacrylic nitric anhydride (Lin et al., 2013; Nguyen et al., 2015), and particle water and other inorganic components. 3-methyltetrahydrofuran-3,4-diols are abbreviated as 3-MeTHF-3,4-diols. Other abbreviations are provided in the main text.





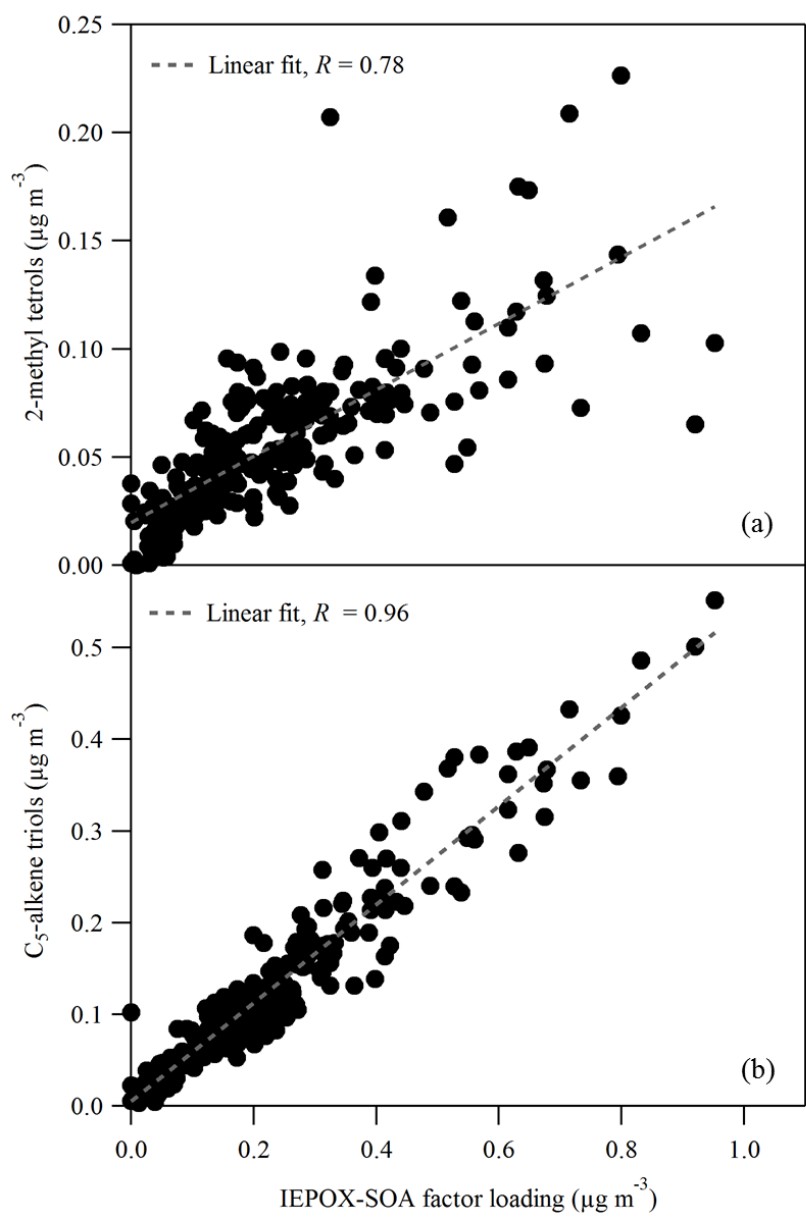

**Figure 2.** Scatter plot of the loading of the IEPOX-SOA factor derived from analysis of the AMS data set and the mass concentrations of $C_5$-alkene triols and 2-methyltetrols measured by SV-TAG. All data collected during IOP1 are included, meaning that the plotted data are not limited to afternoon time periods.





(a) 3 March 2014

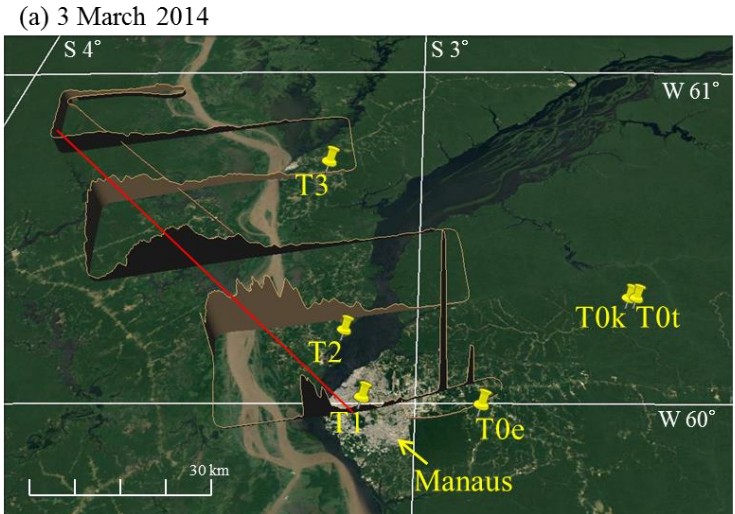

(b) 13 March 2014

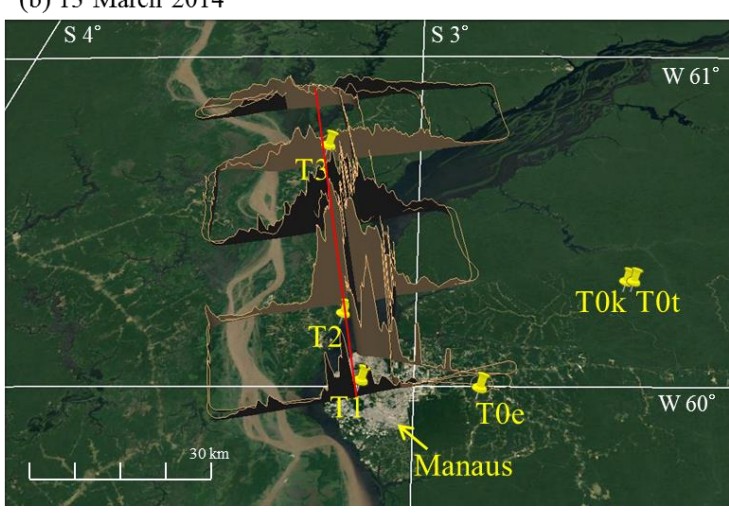

**Figure 3.** Visualization of the Manaus pollution plume by plotting particle number concentrations in the vertical axis. Observations took place on flights from late morning to early afternoon on (a) March 3, 2014, 17:45 – 19:26 UTC, and (b) March 13, 2014, 14:14 – 17:21 UTC. Local time is UTC minus 4 h. The red lines guide the eye through the central axis of the plume. The direction and extent of the plume was observed by the G-1 aircraft within the atmospheric boundary layer downwind of Manaus. Measured particle number concentrations are plotted on a vertical axis on top of an image of land cover in the horizontal plane. Particle concentrations in the center of the plume ranged from 10,000 to 25,000 $cm^{-3}$ nearby Manaus. Yellow pins indicate the locations of some of the GoAmazon2014/5 research sites, including T3 (Martin et al., 2016c).



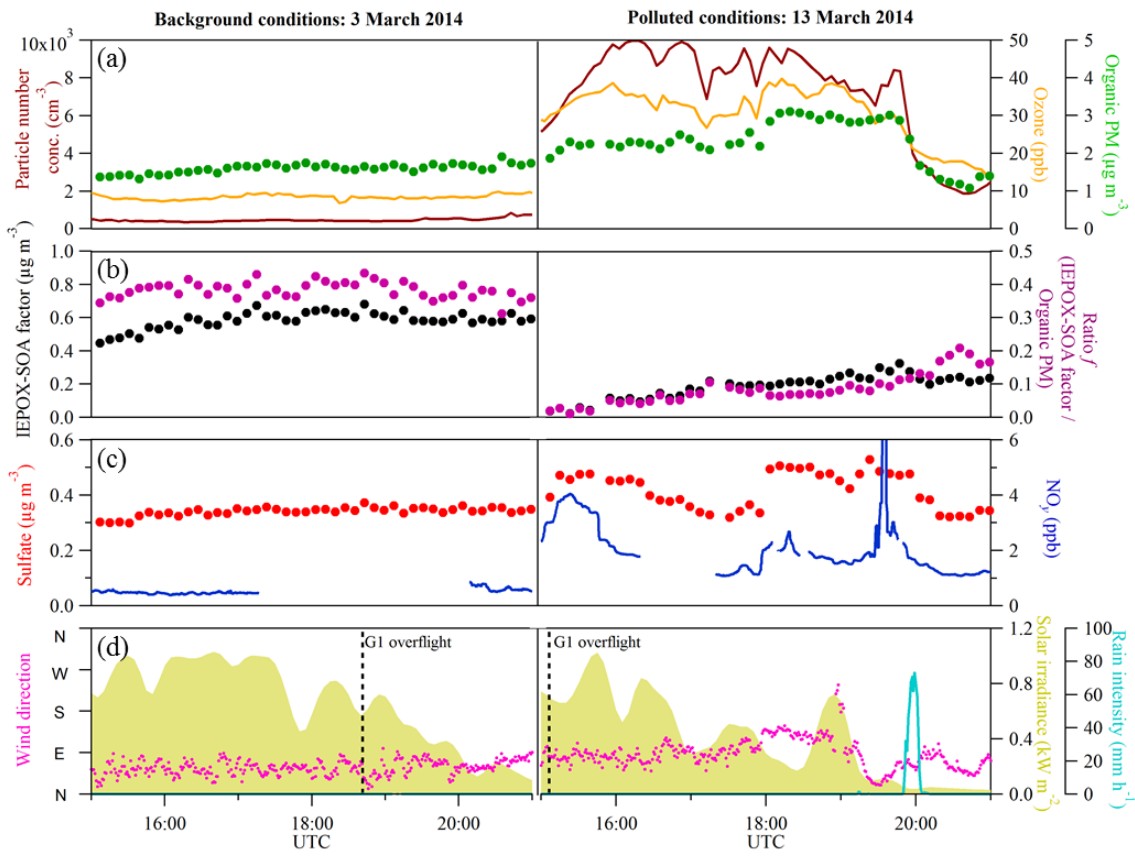

**Figure 4.** Case studies of (left) background and (right) polluted air masses passing over T3 on afternoons of March 3 and 13, 2014. (a) Ozone, particle number, and organic mass concentration. (b) IEPOX-SOA factor loading and the ratio $f$ of the factor loading to the organic PM concentration. (c) Sulfate and $NO_y$ concentrations. (d) Wind direction, rain intensity, and solar irradiance. Local time is UTC minus 4 h. Time points of overflights at 500 m by the G-1 research aircraft are marked by the dashed line (Martin et al., 2016a).





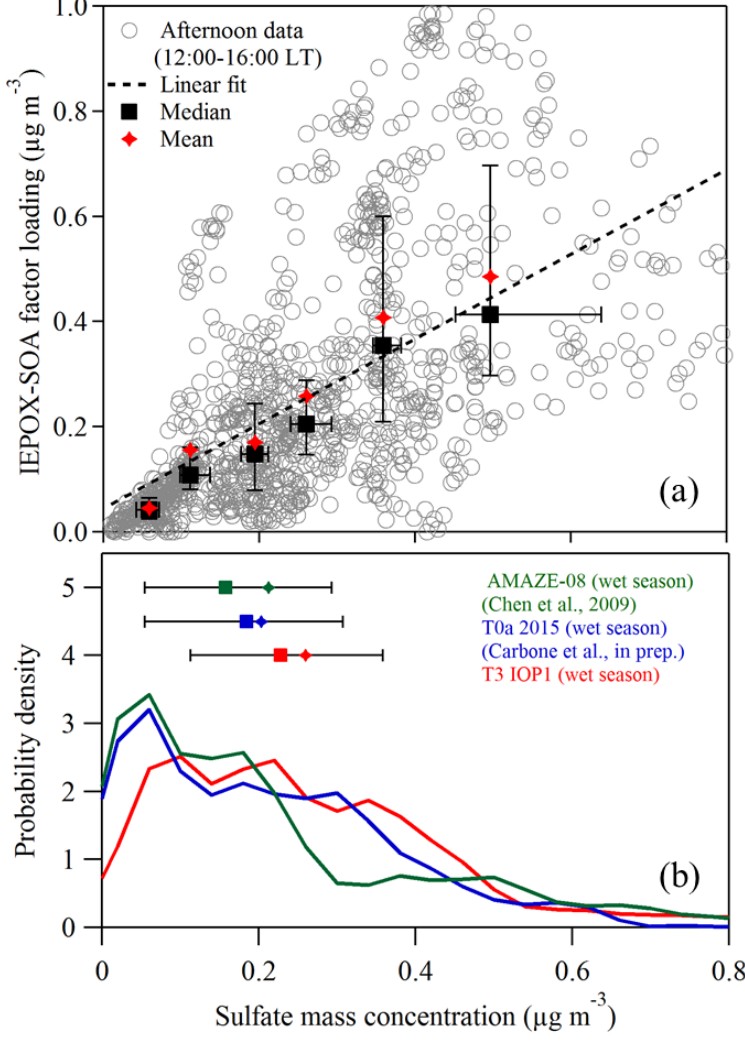

**Figure 5.** (a) Scatter plot of sulfate mass concentration and IEPOX-SOA factor loading. A least-squares linear fit is represented by the dashed line ($R^2$ = 0.37). The data set was collected into six subsets based on sulfate concentration to calculate statistics. Medians (squares) and means (diamonds) of each subset are plotted. Whiskers on the medians represent the interquartile ranges. (b) Probability density function of sulfate mass concentration at the background site T0t ("TT34") north of Manaus in the wet season of 2008 (Chen et al., 2009; Martin et al., 2010b; Martin et al., 2016b), at the background site T0a ("ATTO") northeast of Manaus in the wet season of 2015 (Andreae et al., 2015), and at T3 during the wet season of 2014 (IOP1). The plotted data sets were recorded during local afternoons (12:00-16:00 local time; 16:00-20:00 UTC). Means (diamonds), medians (squares), and interquartile range (whiskers) are shown for the probability density functions.



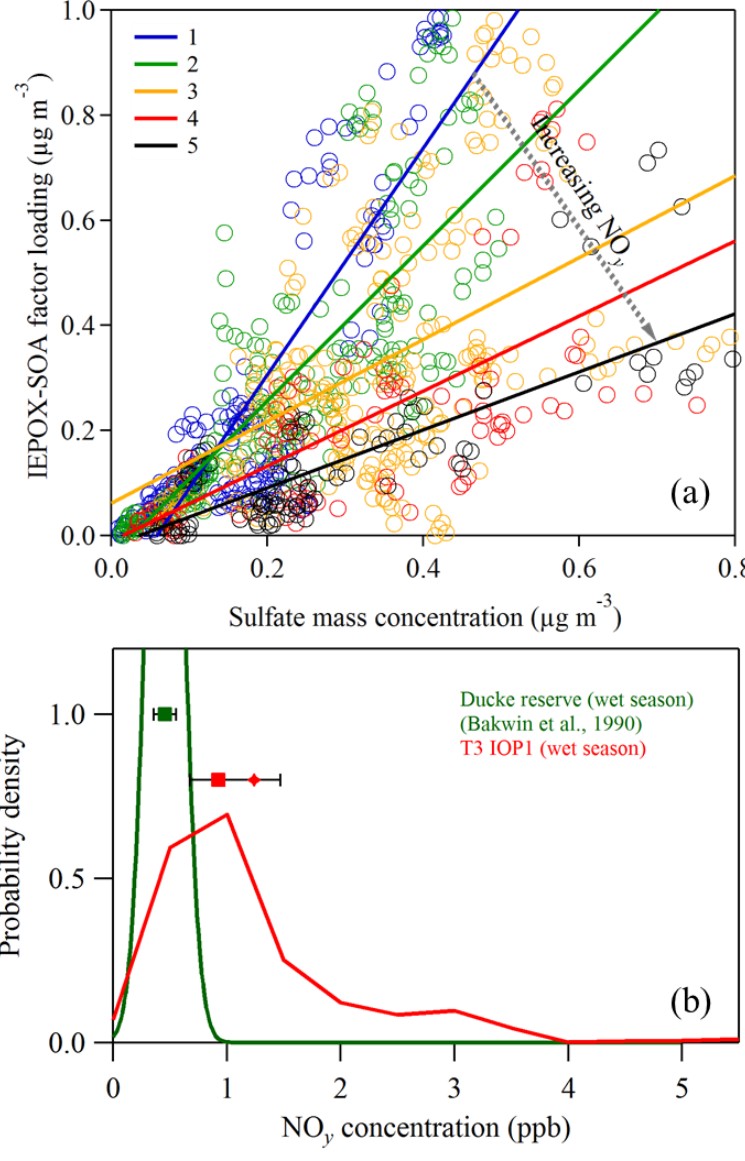

**Figure 6.** (a) Scatter plot of sulfate mass concentration and IEPOX-SOA factor loading for local afternoon (12:00-16:00 local time; 16:00-20:00 UTC). The data sets were collected into five subsets, colored and labeled 1 to 5, based on $NO_y$ concentration. Table 1 presents the parameters of the five least-squares linear fits represented by the colored lines in the figure. (b) Probability density function of $NO_y$ concentration at a background site nearby Manaus in the wet season of 1987 (Bakwin et al., 1990) and at T3 during the wet season of 2014 (IOP1) (afternoon data). Means (diamonds), medians (squares), and interquartile range (whiskers) are shown for the probability density functions. Additional analysis of panel (a) is discussed in the Supplement (Section S5) related to Figure S5.





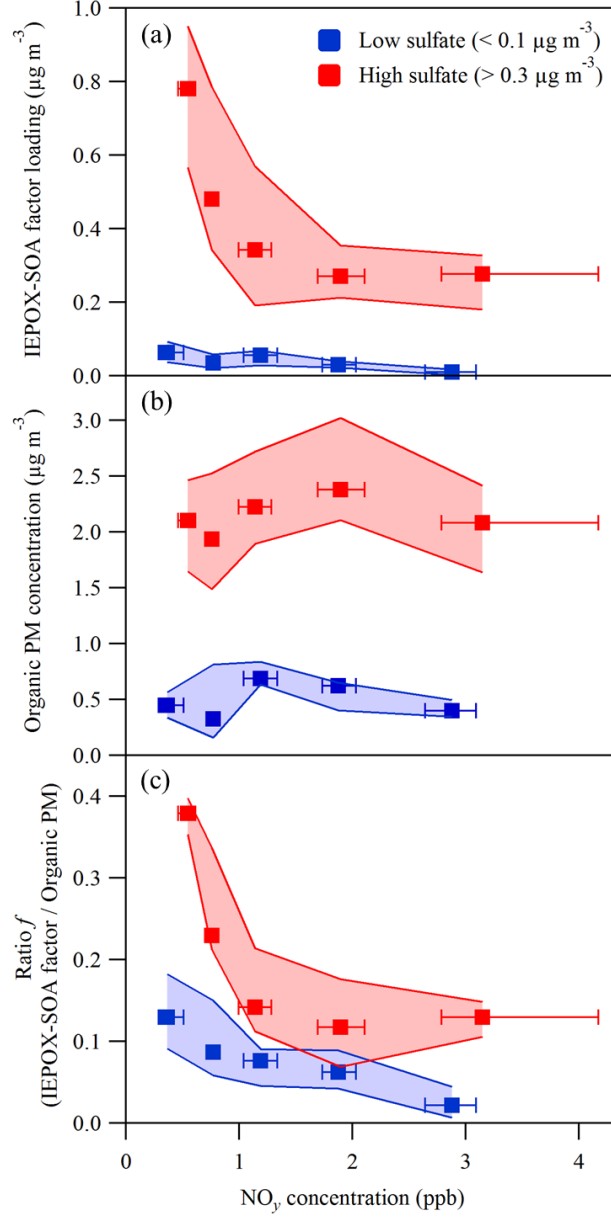

**Figure 7.** Dependence on $NO_y$ concentration of (a) IEPOX-SOA factor loading, (b) organic mass concentration, and (c) the ratio $f$ of the IEPOX-SOA factor loading to the organic PM concentration. Data are segregated by low (< 0.1 µg m$^{-3}$) and high (> 0.3 µg m$^{-3}$) sulfate mass concentration and grouped into five levels of $NO_y$ concentration (Figure 6). Squares represent medians of each group. Interquartile ranges are represented by whiskers along the abscissa and shading along the ordinate. The plotted data sets were recorded during local afternoon (12:00-16:00 local time; 16:00-20:00 UTC).





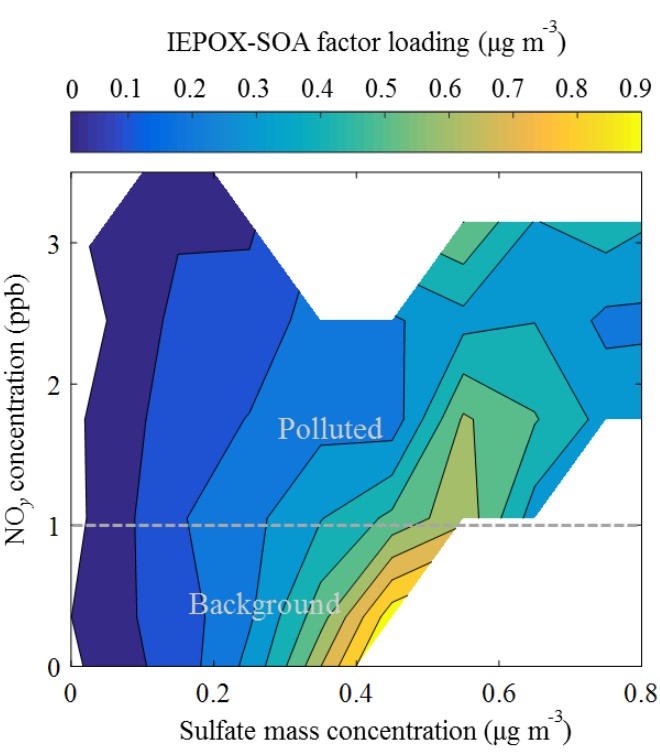

**Figure 8.** Contours of IEPOX-SOA factor loading for sulfate and $NO_y$ concentrations. The plotted data were recorded during local afternoon (12:00-16:00 local time; 16:00-20:00 UTC). Typical transition between regimes of background and polluted conditions for the region downwind of Manaus are approximately represented by the dashed gray line.





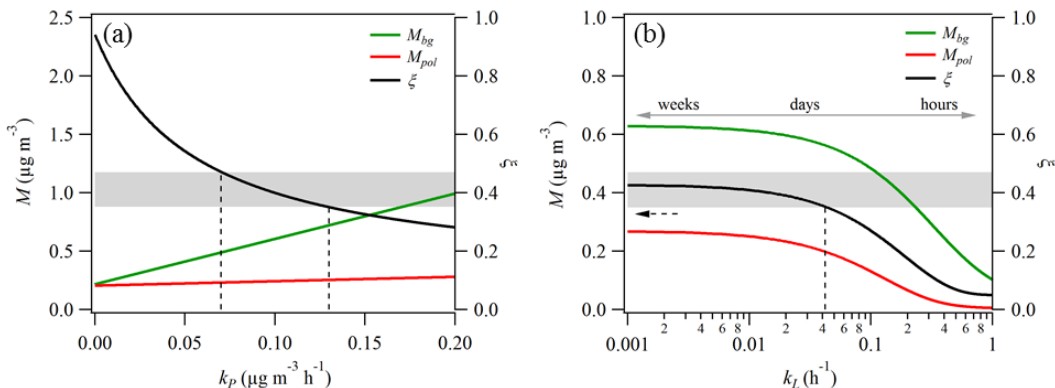

**Figure 9.** Modeled IEPOX-derived PM mass concentrations $M_{pol}$ and $M_{bg}$ at the T3 site under polluted compared to background conditions. The ratio $\xi$ of concentrations (i.e., $M_{pol}/M_{bg}$) is also plotted. Panels a and b correspond to the two model Cases 1 and 2 listed in Table 4 and described in the text. Gray shading indicates the range of observed values of $\xi$ across low and high sulfate concentrations. Dashed lines indicate the intersection of modeled and observed values of $\xi$ and the corresponding constrained values of $k_P$ or $k_L$ along the abscissa. Labels above the double-headed arrow in panel b correspond to characteristic times (i.e., $k_L^{-1}$). The dashed black arrow in panel b communicates that the observed values of $\xi$ provide no constraint on the lower limit of $k_L$.