# Peer review of "Influence of urban pollution on the production of organic particulate matter from isoprene epoxydiols in central Amazonia"

_Atmospheric Chemistry and Physics, 2016_

## Referee Comment (RC1) · Anonymous Referee #1 · 11 Feb 2017

In this work, de Sa et al. presented measurements of isoprene-derived secondary organic aerosol in central Amazonia. Specifically, using positive matrix factorization of aerosol mass spectrometry data, they isolated SOA from isoprene epoxydiol (IEPOX-SOA) and showed the complex dependence of this factor on anthropogenic emissions from nearby urban area (sulfate, nitrogen oxides). This work uses field measurements to highlight some aspects of isoprene chemistry that has been shown first in laboratory and then in other field studies. I believe this work is important and should be published in ACP. I have minor comments only:

1. It is a little puzzling that total PM does not trend with IEPOX SOA. Given the location, I would imagine that in this area the SOA chemistry is dominated by IEPOX SOA. I

understand that there will be another manuscript addressing the full PMF work, but it seems like if there is any location in the world where IEPOX SOA is most dominant, it would be in the Amazon. It seems like the remaining OA dampens some of the variations in IEPOX-SOA, leading to the rather constant total OA. Perhaps the authors can consider just briefly address why in the manuscript?

2. How is the observed sulfate produced? Are they direct emissions from Manaus, or SO2 emitted from Manaus and converted to sulfate in the few hours of transit to the field site? In the conclusions (Ln 534-542) the discussion seems to assume that SO2 and sulfate are equivalent. They might not be if the sulfate is coming from SO2 oxidation in transit.

3. Just another thought about sulfate: could sulfate just be an indicator of degree of oxidation? Since sulfate is a secondary product, it is possible that the variation in sulfate is driven by degree of processing, rather than variation in source strength. Have the authors looked at other indicators of oxidation (e.g. odd oxygen, NOy/NOx ratio, hydrocarbon clocks) to isolate this effect?

4. As the author stated, this environment is different from SE US, in that sulfate levels are generally lower and reduced sulfur can contribute to total sulfate. Are there any indications that reduced sulfur species would be measured as SO4 in the AMS?

5. Are there any estimates of HO2 concentrations? The switch from IEPOX production (under HO2 dominated chemistry) to high NOx chemistry happens at around HO2/NO = 1 (since RO2+NO and RO2+HO2 rate constants are quite close). From the data, it appears that the switch happens at around 0.5 ppb NO, which would suggest that HO2 levels are also around 0.5 ppb. That seems reasonable, but perhaps the authors can explain that in the manuscript to solidify this point.

6. The authors have largely focused on R2 when looking at regressions and demonstrated that, for example, IEPOX-SOA and SO4 are correlated. It would be interesting to also look at the slopes, and compare the sensitivity of IEPOX-SOA to SO4 across the

different field studies (and maybe even lab studies). There are, of course, many other factors that would affect this sensitivity (pH, NOx, aerosol liquid water content etc.). But maybe some simple relationships will emerge that would help construct simplified models to represent these complex chemical systems.

7. In Section 3.1, the authors compared the IEPOX-SOA factor to that observed in other studies. Listing the fractions is useful, but it would be even better to compare the mass spectra (like the authors did with the factor from previous study in the same location). That way it would be more convincing to argue IEPOX-SOA factor is ubiquitous.

—————————————————————

---

## Referee Comment (RC2) · Anonymous Referee #2 · 24 Feb 2017

This is compelling and well written paper, describing observations to elucidate the role of anthropogenic nitrogen oxides and sulfate on the formation of biogenic "IEPOX SOA". The measurements are robust: the identification of IEPOX SOA from PMF factors is grounded in isoprene oxidation tracers. The analysis is carefully considered, and two golden days in which the Manaus plume intercepted the Amazonian field site are used to evaluate the impacts of anthropogenic pollution on the development of SOA over the rainforest. The authors find that IEPOX-SOA increases with sulfate, but once that is controlled for by binning into 'high' versus 'low' sulfate, IEPOX SOA also decreases with NOx exposure. This is consistent with our mechanistic understanding of isoprene oxidation and SOA formation in low NOx environments. Finally, the authors

use a Lagrangian model to demonstrate that the effect of NOx is really to reduce IEPOX SOA production, rather than to increase loss rates. I recommend publication with very minor corrections.

A few points for the authors to consider:

1. The authors describe that sulfate has both background and urban sources, while NOx has just urban sources, complicating the use of sulfate as an anthropogenic tracer (line 374). To what extent do background sources really impact sulfate? (i.e. can the authors quantify this?). I am surprised that sulfate formation from MSA and such would be enough to actually complicate the analysis.

2. The premise of the paper is the relative rate for the ISOPOO radical (an RO2 radical) to react with HO2 versus NO. It may be useful to actually calculate that ratio (so $k_{RO2+HO2}$ [HO2] versus $k_{RO2+NO}$[NO] as a function of NOx. I would anticipate that this ratio maximizes IEPOX SOA formation at the same NOx concentration as the observations show.

Technical comment:

Line 459. I think you mean "model" instead of "mode".

———————————————————

---

## Author Comment (AC1) · 26 Apr 2017

**Response to reviews**

Reviewer comments are in **bold**. Author responses are in plain text. Excerpts from the manuscript are in *italics*. Modifications to the manuscript are in *blue italics*. Page and line numbers in the responses correspond to those in the ACPD paper.

**Review #1**

**In this work, de Sa et al. presented measurements of isoprene-derived secondary organic aerosol in central Amazonia. Specifically, using positive matrix factorization of aerosol mass spectrometry data, they isolated SOA from isoprene epoxydiol (IEPOXSOA) and showed the complex dependence of this factor on anthropogenic emissions from nearby urban area (sulfate, nitrogen oxides). This work uses field measurements to highlight some aspects of isoprene chemistry that has been shown first in laboratory and then in other field studies. I believe this work is important and should be published in ACP. I have minor comments only:**

We appreciate the feedback provided by the reviewer. The revised manuscript takes into account their comments and questions. Detailed responses to each question are given below.

**1. It is a little puzzling that total PM does not trend with IEPOX SOA. Given the location, I would imagine that in this area the SOA chemistry is dominated by IEPOX SOA. I understand that there will be another manuscript addressing the full PMF work, but it seems like if there is any location in the world where IEPOX SOA is most dominant, it would be in the Amazon. It seems like the remaining OA dampens some of the variations in IEPOX-SOA, leading to the rather constant total OA. Perhaps the authors can consider just briefly address why in the manuscript?**

We thank the reviewer for pointing out this aspect. The manuscript has been revised to highlight this point as follows:

Line 415:
*These ranges shifted to [0.35, 0.40] and [0.07, 0.18] for high sulfate concentration.*  *As a limiting statement, for the most favorable conditions with respect to the production of IEPOX-derived PM in central Amazonia (i.e., lowest NO$_y$ and highest sulfate), f exceeded 0.40 at 25% frequency. The implication is that at all times significant additional pathways for PM production were active. This conclusion is subject to the accuracy of the IEPOX-SOA factor loading as a scalar proxy of IEPOX-derived PM concentration (cf. discussion of Figure 1 in Section 3.1). The magnitude of the decrease in f* for high sulfate concentrations suggests that IEPOX-derived PM shifted from being a major to a minor component of the PM.

**2. How is the observed sulfate produced? Are they direct emissions from Manaus, or SO2 emitted from Manaus and converted to sulfate in the few hours of transit to the field site? In the conclusions (Ln 534-542) the discussion seems to assume that SO2 and sulfate are equivalent. They might not be if the sulfate is coming from SO2 oxidation in transit.**

We cannot determine based on our dataset whether sulfate from Manaus is mostly primary or secondary in origin. This point is clarified in the revised manuscript in the following manner:

Line 331:
*The figure shows that the distribution at T3 did not differ greatly from those of the upwind sites even though the air masses over T3 regularly transported Manaus pollution. The implication is that Manaus sulfate sources, whether primary or secondary, had small contributions relative to background sources when averaged over time. In short, elevated sulfate concentrations on any one afternoon at the T3 site might have arisen because of elevated background concentrations on that day rather than the influence of the Manaus pollution plume.*

Line 539*:*
*Based on the findings presented herein, a reduction in sulfate sources from Manaus, whether primary or secondary, would not be expected to considerably affect the mass concentration of IEPOX-derived species in forest regions affected by the plume.*

**3. Just another thought about sulfate: could sulfate just be an indicator of degree of oxidation? Since sulfate is a secondary product, it is possible that the variation in sulfate is driven by degree of processing, rather than variation in source strength. Have the authors looked at other indicators of oxidation (e.g. odd oxygen, NOy/NOx ratio, hydrocarbon clocks) to isolate this effect?**

We appreciate the thoughtful suggestion made by the reviewer. The analysis of indicators of oxidation were complicated by several factors. Absence of $NO_x$, $NO_2$, and $SO_2$ data as well as low signal-to-noise ratio in toluene and benzene measurements precluded a systematic analysis on degree of processing by the several methods mentioned.

Nevertheless, the positive correlation found between IEPOX-SOA factor loadings and sulfate concentrations can be taken as an indication that sulfate variability was not prevalently associated with degree of oxidation. Hu et al. (2015) showed an inverse relationship between $f44$ (the AMS marker for degree of oxidation) and $f82$ (the AMS

marker for IEPOX-SOA). Therefore, if sulfate concentration was mostly an indicator of degree of oxidation and not source strength, IEPOX-SOA factor loading would be expected to decrease with sulfate concentration, which is opposite to what was observed.

**4. a) As the author stated, this environment is different from SE US, in that sulfate levels are generally lower and reduced sulfur can contribute to total sulfate.**

Our intent was to mention reduced sulfur species in the Amazon as precursors to particle sulfate, and not as particle phase components themselves. The text was revised to clarify this point:

Line 316:
*Background concentrations of sulfate in Amazonia, distinguished from sulfate tied to the urban Manaus plume, originated from in-basin emissions of* gas-phase precursors such as *dimethyl sulfide (DMS) and hydrogen sulfide ($H_2S$) from the forest as well as from out-of-basin marine emissions from the Atlantic Ocean (...)*

**b) Are there any indications that reduced sulfur species would be measured as SO4 in the AMS?**

If present as particle phase components, reduced sulfur species could be detected by the AMS (DeWitt et al., 2010). They have not, however, been reported in ambient particles to our knowledge, and we do not see any indication of such reduced sulfur species in the AMS spectra.

**5. Are there any estimates of HO2 concentrations? The switch from IEPOX production (under HO2 dominated chemistry) to high NOx chemistry happens at around HO2/NO = 1 (since RO2+NO and RO2+HO2 rate constants are quite close). From the data, it appears that the switch happens at around 0.5 ppb NO, which would suggest that HO2 levels are also around 0.5 ppb. That seems reasonable, but perhaps the authors can explain that in the manuscript to solidify this point.**

The reviewer raises an excellent question about the gas phase chemistry of isoprene. In fact, Liu et al. (2016a) reported calculated $HO_2$ concentrations at the T3 site and simulated the fate of ISOPOO radicals as a function of NO. The manuscript has been revised as to emphasize the relevance of the findings of Liu et al. (2016a) to this study, as follows.

Line 389:
*The greatest changes in factor loading were in the region of 1 ppb $NO_y$. This region of greatest sensitivity coincided with the transition from background to polluted*

*conditions.*

*For the same time period of these PM analyses of IEPOX-SOA factor loading, Liu et al. (2016a) observed a shift in dominant isoprene gas-phase products from ISOPOOH to MVK/MACR across the transition in NO$_y$ concentration. Liu et al. (2016a) further simulated the dependence on NO concentration of the ratio of the production rate of ISOPOOH to that of MVK + MACR. The highest ratios (0.6 to 0.9) were obtained for background concentrations of NO$_y$. The calculated HO$_2$ concentration was < 4 × 10$^8$ cm$^{-3}$ (0.016 ppb). The simulated transition for the dominant fate of the ISOPOO radicals occurred for an NO concentration of < 0.05 ppb.*

**6. The authors have largely focused on R2 when looking at regressions and demonstrated that, for example, IEPOX-SOA and SO4 are correlated. It would be interesting to also look at the slopes, and compare the sensitivity of IEPOX-SOA to SO4 across the different field studies (and maybe even lab studies). There are, of course, many other factors that would affect this sensitivity (pH, NOx, aerosol liquid water content etc.). But maybe some simple relationships will emerge that would help construct simplified models to represent these complex chemical systems.**

We agree with the reviewer that a comprehensive cross comparison of relationships between IEPOX-SOA and sulfate found in many different studies/locations is valuable, but that is out of the scope of present paper. As demonstrated in this manuscript, even in one location only, the relationship between IEPOX-SOA factor loadings and sulfate concentrations can be dissected by using NO$_y$ concentrations to yield subsets of different slopes (Table 1).

**7. In Section 3.1, the authors compared the IEPOX-SOA factor to that observed in other studies. Listing the fractions is useful, but it would be even better to compare the mass spectra (like the authors did with the factor from previous study in the same location). That way it would be more convincing to argue IEPOX-SOA factor is ubiquitous.**

The reviewer's curiosity is well justified. Hu et al. (2015) addressed the ubiquity and characteristic spectral features of the IEPOX-SOA factor, and the manuscript has been revised as follows.

Line 190:
*(…), and in AMAZE-08 (f = 0.34) (Chen et al., 2015). A further review on the ubiquity and characteristics of the IEPOX-SOA factor is presented in Hu et al. (2015).*

**Review #2**

**This is compelling and well written paper, describing observations to elucidate the role of anthropogenic nitrogen oxides and sulfate on the formation of biogenic "IEPOX SOA". The measurements are robust: the identification of IEPOX SOA from PMF factors is grounded in isoprene oxidation tracers. The analysis is carefully considered, and two golden days in which the Manaus plume intercepted the Amazonian field site are used to evaluate the impacts of anthropogenic pollution on the development of SOA over the rainforest. The authors find that IEPOX-SOA increases with sulfate, but once that is controlled for by binning into 'high' versus 'low' sulfate, IEPOX SOA also decreases with NOx exposure. This is consistent with our mechanistic understanding of isoprene oxidation and SOA formation in low NOx environments. Finally, the authors use a Lagrangian model to demonstrate that the effect of NOx is really to reduce IEPOX SOA production, rather than to increase loss rates. I recommend publication with very minor corrections. A few points for the authors to consider:**

We acknowledge the reviewer for the valuable questions and comments that were provided. These aspects are considered in the revision, and detailed replies to each question are given below.

**8. The authors describe that sulfate has both background and urban sources, while NOx has just urban sources, complicating the use of sulfate as an anthropogenic tracer (line 374). To what extent do background sources really impact sulfate? (i.e. can the authors quantify this?). I am surprised that sulfate formation from MSA and such would be enough to actually complicate the analysis.**

As the reviewer brings up, sulfate sources in the Amazon forest are indeed a very interesting theme. Chen et al. (2009) investigated this topic, and reported that background contributions other than from DMS or $H_2S$ precursors may also be present to a large extent (Section 3.3, lines 316-319). We now see that it would be helpful in Section 3.4 to refer the reader to the relevant section and figures, so we have revised the manuscript as follows.

Line 374:
*In relation to the influence of Manaus pollution, sulfate concentration was affected by a mixture of background and urban sources (cf. discussion in Section 3.3) whereas $NO_y$ concentration largely had urban sources (cf. Figures 5b and 6b).*

**9. The premise of the paper is the relative rate for the ISOPOO radical (an RO2 radical) to react with HO2 versus NO. It may be useful to actually calculate that ratio (so kRO2+HO2 [HO2] versus kRO2+NO[NO] as a function of NOx. I**

**would anticipate that this ratio maximizes IEPOX SOA formation at the same NOx concentration as the observations show.**

The reviewer brings up an excellent point, and we agree that this analysis is important. Fortunately, a co-located study by Liu et al. (2016a) investigated the gas-phase chemistry of isoprene and addressed this question. The manuscript has been revised to include this aspect. Please see answer to comment 5.

**Technical comment:**
**Line 459. I think you mean "model" instead of "mode".**

Our intent here was to refer to the statistical mode of transport time values generated from air mass backtrajectory analysis. This transport time was an input to the model, as opposed to output. To clarify this point, the manuscript was revised as follows.

Line 459:
*The statistical mode value for $\tau_{tr}$ is of 4 h based on trajectory analysis is used in the model (...)*

**References**

Chen, Q., Farmer, D. K., Schneider, J., Zorn, S. R., Heald, C. L., Karl, T. G.,
Guenther, A., Allan, J. D., Robinson, N., Coe, H., Kimmel, J. R., Pauliquevis,
T., Borrmann, S., Pöschl, U., Andreae, M. O., Artaxo, P., Jimenez, J. L., and
Martin, S. T.: Mass spectral characterization of submicron biogenic organic
particles in the Amazon Basin, Geophys. Res. Lett., 36, L20806, 2009,
10.1029/2009GL039880.

Chen, Q., Farmer, D. K., Rizzo, L. V., Pauliquevis, T., Kuwata, M., Karl, T. G.,
Guenther, A., Allan, J. D., Coe, H., Andreae, M. O., Pöschl, U., Jimenez, J. L.,
Artaxo, P., and Martin, S. T.: Submicron particle mass concentrations and
sources in the Amazonian wet season (AMAZE-08), Atmos. Chem. Phys., 15,
3687-3701, 2015, 10.5194/acp-15-3687-2015.

DeWitt, H.L., Hasenkopf, C.A., Trainer, M.G., Farmer, D.K., Jimenez, J.L., McKay,
C.P., Toon, O.B., and Tolbert, M.A.: The formation of sulfate and elemental
sulfur aerosols under varying laboratory conditions: implications for early
Earth, Astrobiology, 10, 773-781, 2010, 10.1089/ast.2009.9455.

Hu, W. W., Campuzano-Jost, P., Palm, B. B., Day, D. A., Ortega, A. M., Hayes, P. L.,
Krechmer, J. E., Chen, Q., Kuwata, M., Liu, Y. J., de Sá, S. S., McKinney, K.,
Martin, S. T., Hu, M., Budisulistiorini, S. H., Riva, M., Surratt, J. D., St. Clair,
J. M., Isaacman-Van Wertz, G., Yee, L. D., Goldstein, A. H., Carbone, S.,
Brito, J., Artaxo, P., de Gouw, J. A., Koss, A., Wisthaler, A., Mikoviny, T.,
Karl, T., Kaser, L., Jud, W., Hansel, A., Docherty, K. S., Alexander, M. L.,
Robinson, N. H., Coe, H., Allan, J. D., Canagaratna, M. R., Paulot, F., and

Jimenez, J. L.: Characterization of a real-time tracer for isoprene epoxydiolsderived secondary organic aerosol (IEPOX-SOA) from aerosol mass

spectrometer measurements, Atmos. Chem. Phys., 15, 11807-11833, 2015,

10.5194/acp-15-11807-2015.

Liu, Y., Brito, J., Dorris, M. R., Rivera-Rios, J. C., Seco, R., Bates, K. H., Artaxo, P.,

Duvoisin, S., Keutsch, F. N., Kim, S., Goldstein, A. H., Guenther, A. B.,

Manzi, A. O., Souza, R. A. F., Springston, S. R., Watson, T. B., McKinney, K.

A., and Martin, S. T.: Isoprene photochemistry over the Amazon rain forest,

Proc. Natl. Acad. Sci. USA, 113, 6125-6130, 2016a,

10.1073/pnas.1524136113.

---

## Author Comment (AC2) · 26 Apr 2017

The comment was uploaded in the form of a supplement:
http://www.atmos-chem-phys-discuss.net/acp-2016-1020/acp-2016-1020-AC2-supplement.pdf